# An Inspection and Classification System for Automotive Component Remanufacturing Industry Based on Ensemble Learning

**Fátima A. Saiz** [1,2,*] **, Garazi Alfaro** [1] **and Iñigo Barandiaran** [1]

1   Vicomtech Foundation, Basque Research and Technology Alliance (BRTA), Mikeletegi 57,
    20009 Donostia-San Sebastián, Spain; galfaro@vicomtech.org (G.A.); ibarandiaran@vicomtech.org (I.B.)
2   Computational Intelligence Group, Computer Science Faculty, University of the Basque Country (UPV/EHU),
    20018 Donostia-San Sebastian, Spain
*   Correspondence: fsaiz@vicomtech.org

**Abstract:** This paper presents an automated inspection and classification system for automotive component remanufacturing industry, based on ensemble learning. The system is based on different stages allowing to classify the components as good, rectifiable or rejection according to the manufacturer criteria. A study of two deep learning-based models' performance when used individually and when using an ensemble of them is carried out, obtaining an improvement of 7% in accuracy in the ensemble. The results of the test set demonstrate the successful performance of the system in terms of component classification.

**Keywords:** quality inspection; deep learning; ensemble learning; component remanufacturing; automotive industry

## 1. Introduction

Facing the emerging environmental crisis that the planet is experiencing, it is in everyone's hands to implement urgent actions that help prevent climate change and promote both sustainable development and environmental protection. From an industrial perspective, one of the lines of action that can be taken to achieve sustainable manufacturing is remanufacturing [1–3].

### 1.1. Remanufacturing Process in the Manufacturing Industry

According to [4], remanufacturing could be defined as a process of returning used products to a functional "as new" state by rebuilding and replacing their components. This process is an effective way to reduce emissions of carbon dioxide ($CO_2$) and other greenhouse and global warming gases [5]. Remanufacturing also reduces skilled labour, energy, material waste and overexploitation of natural resources, among others benefits [6–9]. In practice, companies develop remanufacturing businesses for social, economic and environmental benefits [10,11].

Remanufacturing is the reuse of products that have reached the end of their usable life, by carrying out a series of processes that return the product to its original state with an equivalent or superior quality. Therefore, the warranty of the remanufactured product is identical to the warranty of a new product. This process is also environmentally friendly because reduces energy consumption by eliminating the need to produce new components. In addition, it can significantly reduce lead times, thereby increasing customer satisfaction.

A typical remanufacturing process can be seen in Figure 1. This cyclical process starts with the availability of a used part (core); initially, this part was manufactured in a linear manufacturing process that complies with specific technical specifications. In the case of products with different components, the product is completely disassembled and an

assessment of the condition of each individual part is carried out through an inspection system. Once the quality status of the component is defined, it is classified according to whether it is usable, it needs an additional repairing process or it is unusable. The parts classified as usable or repairable are subjected to an intensive cleaning process. The parts are then repaired and upgraded as necessary, where they are subjected to a series of advanced manufacturing processes. The cleaned, repaired and processed parts are then reassembled to make the final product. Finally, the reconditioned product are tested and evaluated to ensure that its condition meets the conditions and technical specifications of new products.

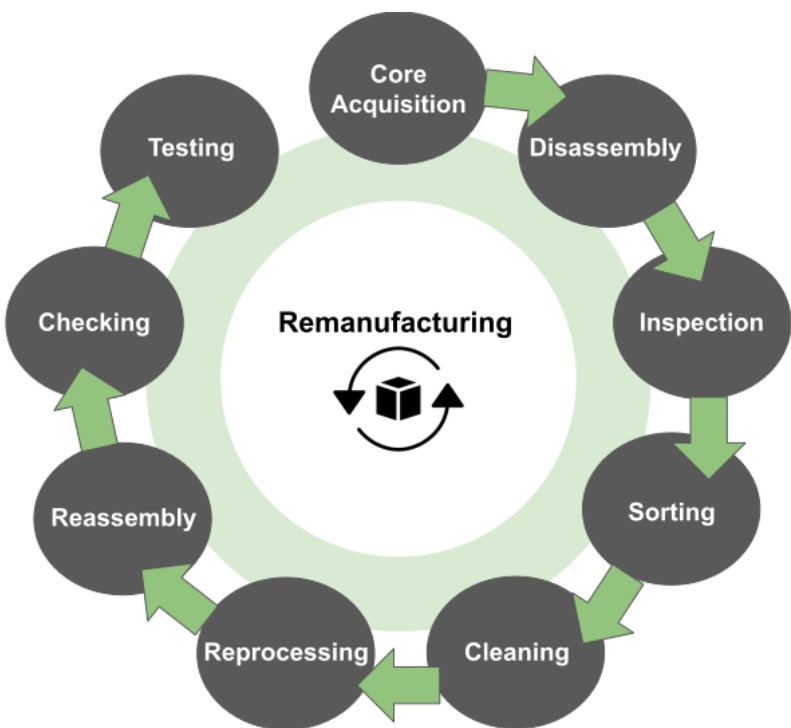

**Figure 1.** Example of remanufacturing process.

There are various sectors that can benefit from the remanufacturing process [12], such as aerospace, automotive [13] or electronics. Although this process is experiencing growth, some of this growth is constrained by factors such as the complex nature of the remanufacturing process [14]. There are other constraints such as the difficulty in obtaining an adequate supply of used products, as well as the development of efficient remanufacturing process tools and techniques to carry out the previously mentioned different steps of a remanufacturing process. The remanufacturing industry is therefore facing a major challenge to achieve a more sustainable production, which is so necessary to tackle the environmental crisis.

### 1.2. Machine Vision Applications for Quality Control

Among the different processes that compose the remanufacturing cycle, this paper is focused on the inspection phase. Inspection is absolutely necessary to determine the degree of deterioration and the quality of the product in order to facilitate the classification between good parts, rectifiable parts and rejectable parts that cannot be remanufactured. In this work, we address a real industrial use case of an inspection system focused on the remanufacturing of an automotive mechanical component.

Traditionally, inspection is performed manually by an operator. To overcome the limitations of human inspection, such as time, high labour cost or individual subjectivity, automated inspection techniques have started to be implemented to assist or replace human decisions. In recent years, methods based on deep learning and computer vision

have achieved excellent performance on automated visual inspection problems [15–18]. Through these neural models, data acquired in production environments can be analysed and learned, with the aim of enhancing the inspection process with human-like skills. As a result, visual inspection has changed from being carried out manually by an operator to being fully automated.

With increasingly demanding quality standards, quality control is a major challenge for the industrial manufacturing sector. The need for strict quality inspection in remanufacturing is critical, as the final quality of the component depends on this process. In addition, achieving a good inspection performance in the imposed processing time is a difficult task that requires new neural networks architectures [19]. The studies that are currently being carried out on multiple types of components such as metal sheets [20,21], plastic pipes [22] or metallic brackets [23], among others, show the suitability of deep learning techniques for the inspection stage. The main problem with deep learning methods is that they generally require an inspection of the prediction results that do not have a high confidence value. In order to achieve more stable models that are able to specialize in complex image features, model ensembles are used [24,25]. This strategy allows each individual model to learn certain details that the rest of the models do not have to learn, thus parallelizing the detection tasks and obtaining multiple detection "opinions".

### 1.3. Main Contributions

We propose to use a deep learning model ensemble for the inspection and classification of constant velocity joint cages. We propose to combine these approaches in a system that detects whether the cages have a wear (defect) or not, and then classifies them as rejectable or rectifiable, according to the manufacturer's criteria about their defect size. We demonstrate the benefit of this combination compared with the use of each model in isolation.

The structure of this paper is as follows: In Section 2, the characteristics of the inspected components are described, as well as our inspection and evaluation proposed pipeline. In Section 3, the results of the experimentation are shown, followed by a discussion. Finally, in Section 4, some conclusions and possibilities for future work are presented.

## 2. Materials and Methods

This section describes in more detail the characteristics of the components that are analysed in this case study, together with their defectology definition. In addition, the proposed inspection and evaluation pipeline is described. Finally, the metrics to be used in the system evaluation stage are also detailed.

### 2.1. Characteristics of Inspected Components

In this paper we are going to use as a case study a component used in the automotive sector. Specifically, the study focuses on the Constant Velocity (CV) joints. The CV joint is a mechanical articulation in which the rotational speed of the output shaft is the same as that of the input shaft, regardless of the transmission angle at which the joint operates. Its design allows the rotational motion to be transmitted through cross grooves located between an outer bell and a grooved inner. The most commonly used design today, to which our components belong, is the Rzeppa type [26]. In this design, the balls are held in position by small windows in a mounting cage between the outer bell and the inner.

The design of the joint is such that the position of the balls always bisects the operating angle of the joint. It is a design that works like a ball gear. These balls cause wear in the area of contact with the cage, creating small marks in the part. In this paper we refer to this defect as wear. Figure 2 shows an image of the bearing contact zone, that is the region to be inspected.

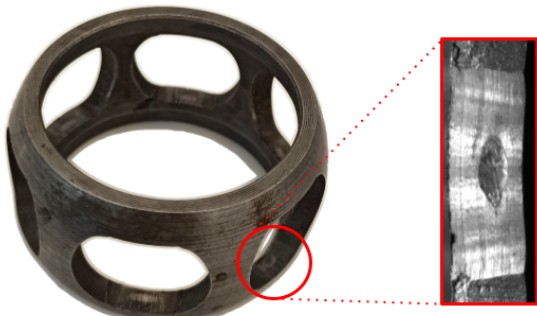

**Figure 2.** Remanufactured automotive component (CV joint cage).

The size of the wear is a discriminating factor in assessing whether the part can be reused again, must be rectified or it is no repairable. In this case, the criterion set by the manufacturer is as follows:

- If the cage has a wear diameter smaller than 0.25 mm, it is rectifiable;
- If the cage has a wear diameter equal or greater than 0.25 mm, it is rejectable.

There are different models of cages which are characterised, among other things, by the number and shape of their bearing contact points and windows. Figure 3 shows an example of the variability shown by the bearing contact zone in the image acquisition. The wear is highlighted by the red areas with a dashed border.

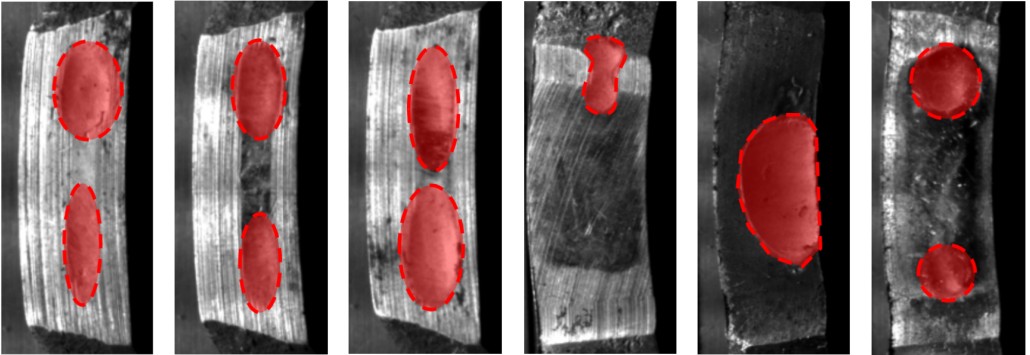

**Figure 3.** Some examples of the dataset of remanufactured parts with different degrees of surface wear coloured in red with a dashed border.

*2.2. Proposed Inspection and Evaluation Pipeline*

The proposed pipeline is shown in Figure 4 and explained in detail in the following sections.

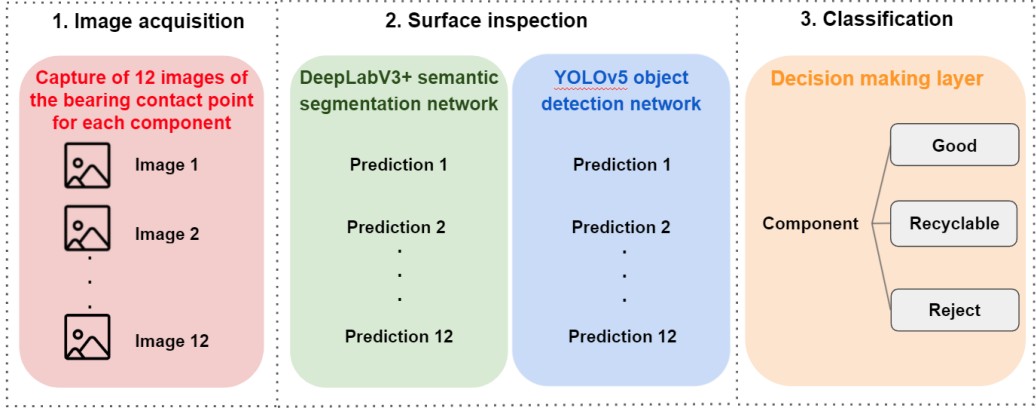

**Figure 4.** Proposed inspection and evaluation pipeline steps.

### 2.2.1. Step 1: Image Acquisition

The first step in the proposed pipeline is to acquire the component images. The proposed acquisition system aims to highlight as much as possible the wear in the image. It is composed of the following elements as shown in Figure 5: a 5MP monochromatic matrix camera, a lighting bar oblique to the inspection area to maximise the contrast, and a centring device to place the cage inside the camera field of view. Thanks to this configuration, the obtained images have shown high contrast in the wear, regardless of the geometrical characteristics and the polishing level of the bearing contact zone. An example of acquired images with this setup is shown in Figure 3.

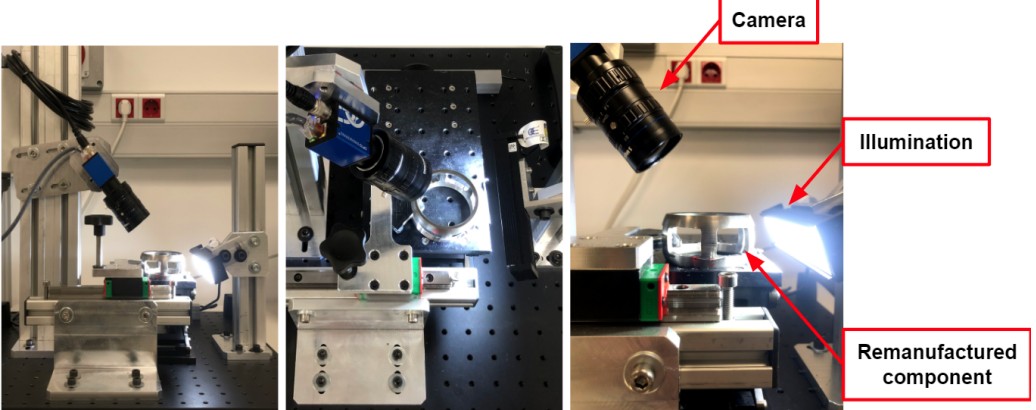

**Figure 5.** Image acquisition setup.

### 2.2.2. Step 2: Surface Inspection

Our proposal is to use artificial intelligence-based mechanisms for surface inspection. Specifically, the used neural networks in this work are: DeepLabV3+ [27] and YOLOv5 [28]. DeepLabV3+ is a semantic segmentation network with a decoder that improves the segmentation results with respect to its predecessor, DeepLabV3. Thanks to the downsampling, the resolution of the feature maps is reduced. This network has the peculiarity of using Atrous Convolutions, which allow to refine the effective field of view of the convolution. The result of this network is a mask, which will allow us to obtain an estimation of the wear dimension. The architecture of DeepLabV3+ is shown in Figure 6.

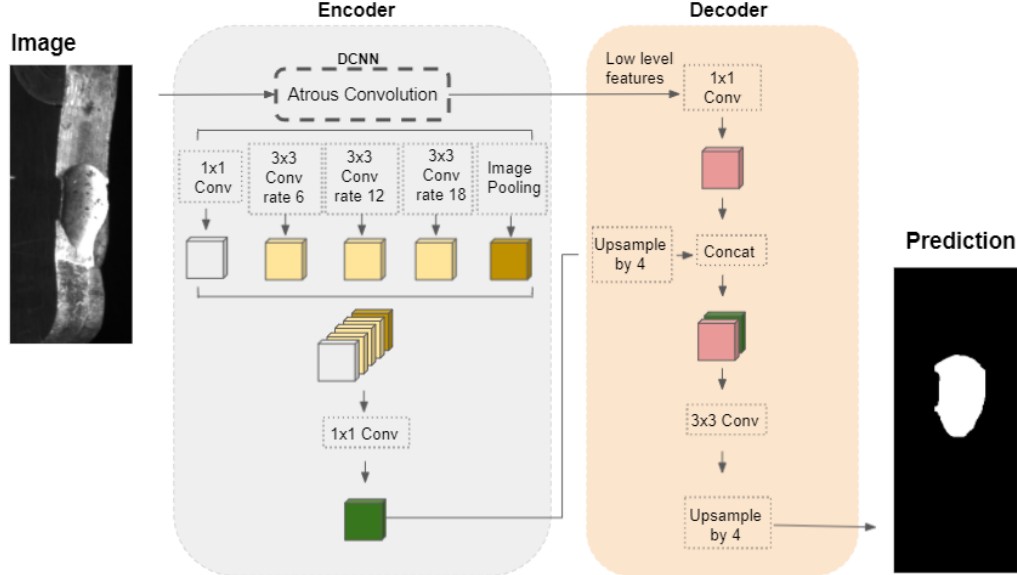

**Figure 6.** The network architecture of DeepLabV3+.

The second network used is YOLOv5. YOLOv5 is an object detection network, whose strategy is to divide the image into a grid, where objects are detected in each grid. In terms of architecture, this network is divided into three main blocks: the BackBone, the neck and the yolo head, as shown in Figure 7. The backbone of YOLOv5 is CSPDarknet [29] which employs a CSPNet strategy to partition the feature map of the base layer into two parts and then merges them through a cross-stage hierarchy. The use of a split and merge strategy allows for more gradient flow through the network. The neck, PANet [30] allows all the features to be merged. Finally, at the head is the yolo layer, which presents the output results. The major improvements of this yolo version is that it includes mosaic data augmentation and auto learning bounding box anchors.

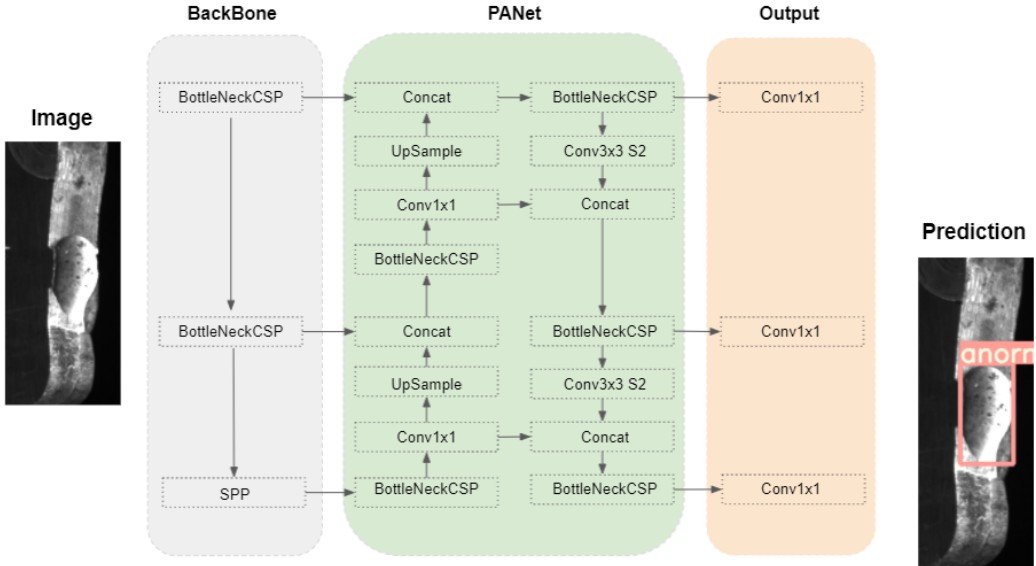

**Figure 7.** The network architecture of YOLOv5.

### 2.2.3. Step 3: Classification Layer

The last step of the pipeline is the decision layer. The purpose of this layer is to merge the outputs of the surface inspection stage. In this layer is integrated all the decision logic required by the manufacturer. The final result is a single final classification in which all the models used are involved in the decision. For this purpose, decision trees are used, which are commonly used in decision making based on a series of conditions that happen in a successive way [31].

### 2.3. Evaluation Metrics

Different metrics are used to evaluate the models: Intersection over Union (IoU), mean Average Precision (mAP) and the accuracy.

A common way to determine whether a prediction proposal is correct is to use Intersection over Union (IoU) [32]. The IoU specifies the amount of overlap of the bounding boxes or pixels between the prediction and the ground truth, in a value from 0 to 1, being better when is closer to 1. The equation to calculate the IoU is shown in Equation (1), where $A$ is a pixel set of proposed object and $B$ the pixel set of ground truth object.

$$IoU(A, B) = \frac{A \cap B}{A \cup B} \tag{1}$$

Based on the IoU result obtained, we classify whether the prediction is a False Positive, True Positive, False Negative or True Negative. Where this means:

- True Positives (TP): the defect is detected as defect;
- True Negatives (TN): the normality is detected as normality;

- False Positives (FP): the normality is mistakenly detected as defect;
- False Negatives (FN): the defect is mistakenly detected as normality.

Traditionally, a prediction is classified as TP if the IoU is >0.5. With the obtained values of TP, TN, FP and FN the accuracy and recall using the Equations (2) and (3), respectively, is calculated.

$$\text{Precision} = \frac{\text{TP}}{\text{TP} + \text{FP}} \tag{2}$$

$$\text{Recall} = \frac{\text{TP}}{\text{TP} + \text{FN}} \tag{3}$$

The Precision and Recall values are used to create the PR curve, where the Precision is plotted on the Y-axis and the recall on the X-axis. From this curve the AP can be calculated, whose value is the area under the PR curve. The mAP for object detection is the average of the AP value calculated for all classes.

For the classification evaluation, at the individual image level and at the cage level, the accuracy is used. The accuracy in binary classification, as in this case, is calculated in terms of positives and negatives. Formally, accuracy has the definition shown in Equation (4), where:

$$\text{Accuracy} = \frac{\text{TP} + \text{TN}}{(\text{TP} + \text{TN} + \text{FP} + \text{FN})} \tag{4}$$

## 3. Results and Discussion

The experimentation of this work is composed of different tests that validate which inspection method is the most suitable for the classification of CV joint cages. This experimentation covers different aspects such as the performance of each model used individually and the benefits obtained by networks ensemble.

### 3.1. Dataset Generation

The dataset generation is carried out using the data acquisition system described in Section 2.2.1. A total of 55 CV joint cages are acquired, where 12 bearing contact point images are taken for each component. Summarizing, the complete dataset has 660 images. As shown in Table 1, the database is divided into three different sets, a training set, a validation set and a test set.

**Table 1.** Distribution of the database for training and testing of DeepLabV3+ and Yolov5 models.

|  | Total Set | Training Set | Validation Set | Test Set |
|---|---|---|---|---|
| Number of remanufactured components | 55 | 36 | 9 | 10 |
| Number of images (12 wear zone per component) | 660 | 432 | 108 | 120 |

In terms of the quality of the training set data, a key factor that has a direct impact on the performance of the neural network is that the set is well-balanced. Especially in manufacturing environments, it is very common to have a shortage of defective data. A commonly applied technique to overcome this shortage and make models more robust is to apply data augmentation. This technique consists of increasing the volume of the training set, based on a series of geometric and photometric operations. In this case, the operations applied are rotation, incorporation of Gaussian noise, flipping effect, mirror effect and brightness variation. The processing operations must resemble the conditions of the working environment in which the image is acquired. Thus, a 15-fold increase in the original data volume was realized, from 432 images opted for training increased to a total of 6480 samples.

### 3.2. Performance Comparison between Traditional Methods and Deep Neural Networks

In this experiment, an evaluation with different traditional machine learning methods was carried out with the test set. Although in this paper a surface inspection based on deep learning is proposed, some classical machine learning techniques such as SVM (Support Vector Machine) [33], Gaussian Naive Bayes [34] and decision trees [35,36] are compared. The objective of this experiment is to conclude whether the use of deep learning is really essential for a surface inspection that is robust and adaptable to extreme production environments and also complies with the strict quality control standards set by the manufacturing industry.

The result obtained with all the traditional methods are very similar, as shown in the performance metrics shown in Table 2. It is observed that there is a strong trend to the faulty prediction of the sample as there is a significant number of false positives (FP). Similarly, by observing the values of the most classical methods metrics, it shows that they make a practically random inspection for the non-faulty samples.

**Table 2.** Performance metrics of evaluated methods using the test set.

| Method | TP | TN | FP | FN |
|---|---|---|---|---|
| Decision Tree | 46 | 20 | 32 | 22 |
| Gaussian Naive Bayes | 54 | 28 | 24 | 14 |
| SVM | 59 | 24 | 28 | 9 |
| DeepLabV3+ | 51 | 50 | 2 | 17 |
| UNet | 50 | 48 | 13 | 9 |
| YOLOv3 | 53 | 36 | 10 | 21 |
| YOLOv5 | 60 | 44 | 8 | 8 |
| **YOLOv5+DeepLabV3+** | **62** | **50** | **2** | **6** |

Thanks to experimentation with different traditional machine learning methods, it is shown that, in comparison, deep learning-based models, such as Unet [37], DeepLabV3+, YOLOv3 [38] and YOLOv5, yield much better results. It could even be observed that the classical methods fail in learning the defects, which addresses the surface inspection problem, due to the large number of errors in form of false positives (FP) they make during the evaluation. FPs generate failures in the detection entailing an over-detection of defects. This means that the model does not learn the characteristics of the defects properly. In industry, this implies a very large over-rejection, which raises doubts about the reliability of the model in an automatic inspection process, as it would mean an extra cost due to the waste of good parts. Therefore, we can state that the use of more complex deep learning architectures, such as Unet, DeepLabV3+, YOLOv3 and YOLOv5, is needed to achieve high inspection accuracy rates.

### 3.3. Individual Evaluation of the Deep Neural Networks Models Performance

Based on the defect detection results obtained in the previous experiment using different deep neural network models, we selected the best segmentation and object detection models, DeepLabV3+ and YOLOv5, respectively. In this experimentation two trainings were performed, one with the semantic segmentation model DeepLabV3+ and the other with the defect detection model YOLOv5. For both neural models, the same dataset defined previously in Table 1 was used.

During the training of the DeepLabV3+ and YOLOv5 models an evaluation was performed for each epoch with the validation set in order to have a feedback of the training and to ensure its convergence to obtain satisfactory results.

The evaluation of each model was performed with the test set. As mentioned in the Section 2.3, different evaluation metrics such as IoU, mAP and Accuracy were used. The YOLOv5 network achieved 90% of mAP and the DeepLabV3+ network 85% of IoU. Both models yielded excellent results in terms of surface wear detection.

Some predictions of the two models are shown in Figure 8. This figure shows the semantic segmentation predicted masks using DeepLabV3+ in Figure 8b, as well as the predicted bounding boxes of YOLOv5 Figure 8c. The masks and bounding boxes are shown in red over the defective regions, and can be observed that they perfectly match the contour of the wear zone, achieving outstanding inspection accuracy.

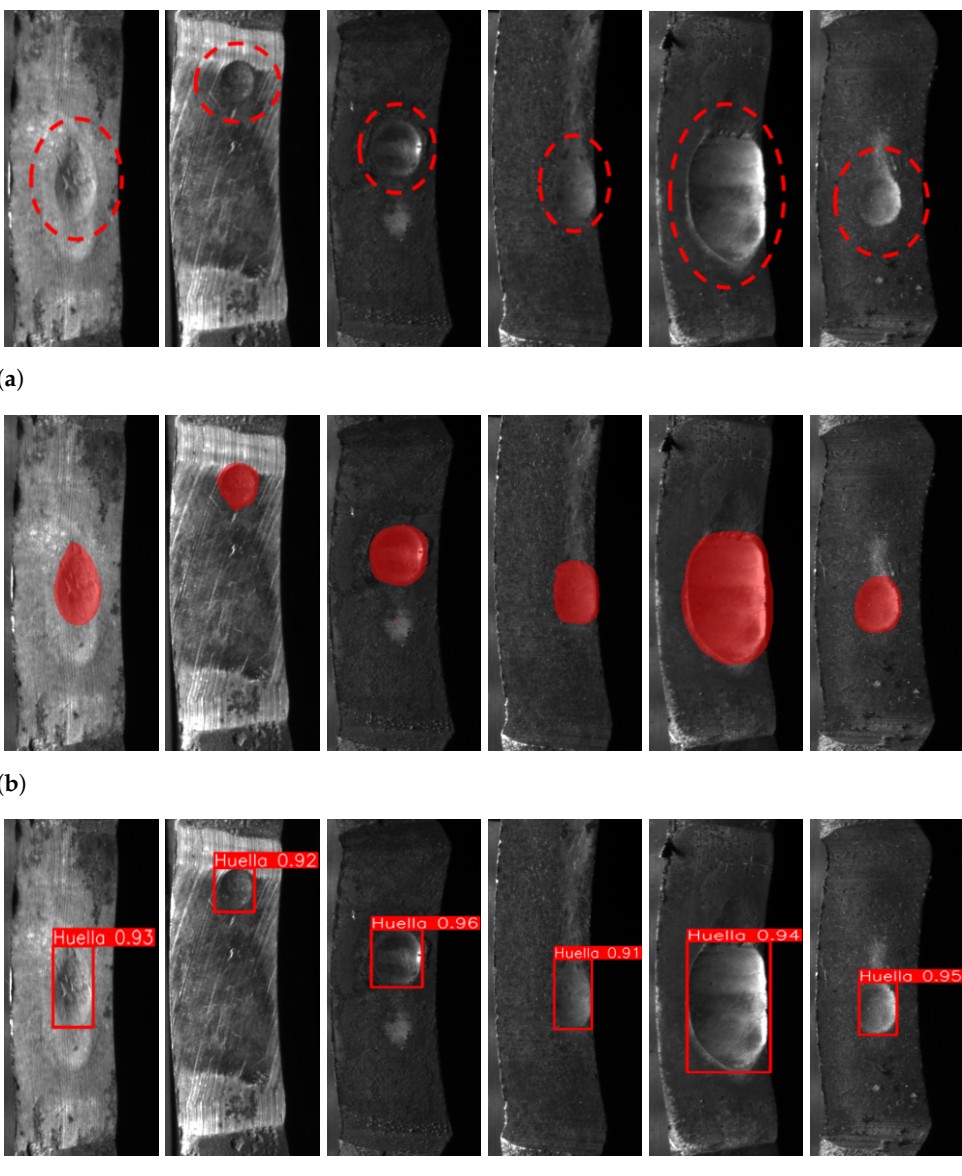

(**a**)

(**b**)

(**c**)

**Figure 8.** Surface wear detection using DeeplabV3+ semantic segmentation network and YOLOv5 detection network individually; (**a**) CV joint cage bearing contact point images and the wear zone indicated by red dotted lines, (**b**) prediction of semantic segmentation of the DeepLabV3+ model shown in red and (**c**) prediction of YOLOv5 model depicted with red bounding box.

During the experimentation it was observed that the models have different performance and capabilities in terms of surface inspection. The YOLOv5 network shows higher sensitivity for detecting wear regions, thus achieving better results in the evaluation step. However, it is prone to have false positives (FP), which leads to a decrease in detection reliability. In contrast, the DeepLabV3+ network gains in robustness, as it has almost no false positives (FP) in the detection, meaning that the defects detected are actually defective areas. DeepLabV3+ has more false negatives (FN) than YOLOv5, indicating a slightly lower accuracy in the evaluation metrics, as shown in Table 3.

**Table 3.** Results of the Accuracy metric from the evaluation phase of the different YOLOv5 and DeepLabV3+ neural models.

| Model | DeepLabV3+ | YOLOv5 | YOLOv5 + DeepLabV3+ |
|---|---|---|---|
| Accuracy (%) | 84.17 | 86.67 | 93.33 |

### 3.4. Analysis of the Model Ensemble Performance

In this experiment, the DeepLabV3+ and YOLOv5 models are assembled to obtain a single class output. For this purpose, a decision layer is developed where the outputs of both networks are combined, the wear of each bearing contact point is measured in millimetres and the classification of the CV joint cage is performed.

Both models, DeepLabV3+ and YOLOv5, gave excellent results in the individual evaluation on the test set. However, it was observed that the two models shows different performance for the inspection of the addressed surface. While YOLOv5 has a trend to detect practically all wear zones, it has many false positives (FP) in the inspection, although it deals better with intra-class variability. In contrast, DeepLabV3+ is a model that has more false negatives (FN) than YOLOv5, but the detections are more confident and it is able to find very small wear zones.

Therefore, the YOLOv5 and DeepLabV3+ models are combined in order to reduce the mistakes produced in the inspection. In Table 3, the classification result during the evaluation phase of the YOLOv5+DeepLabV3+ ensemble is shown. The ensemble obtained an accuracy metric of 93.33%, a higher value compared with the models evaluated individually. In Figure 9, some results obtained with the YOLOv5+DeepLabV3+ ensemble are shown, where the wear zones of the bearing contact points are highlighted with a red bounding box. In Figure 9b, it can be observed how in some samples the surface wear was detected by only one model while the other did not detect it, when using both models isolated, i.e., not using the ensemble. In these cases, the YOLOv5+DeepLabV3+ ensemble is an effective method to avoid errors in the inspection, supplying the shortcomings shown by the two models individually. Therefore, the potential of merging more than one neural network to tackle a complicated surface inspection problem, where a database with a lot of intra-class variability is used, is validated.

### 3.5. Component Final Classification Results

In this last experiment, the ability of the system to classify components as rejectable, rectifiable or valid is evaluated. In order to classify a component into the mentioned three categories, it is necessary to inspect the 12 bearing contact points of each component individually. This classification by contact point is performed by using the assembled model and considering several quality criteria defined by the manufacturer. To make this classification, a decision tree is proposed as shown in Figure 10. Thanks to this tree, the class predicted by each model is obtained, together with its weight, which was defined based on the individual performance of the models. With these data, a weighted average is calculated to obtain the final class for each bearing contact point image.

Component-level classification evaluation was performed using a test set composed of 10 different cages. The criterion used to classify a component as defective is that it has more than four wears detected in total. This criterion is defined by the manufacturer due to the geometry of the cage, since, from a geometrical and functional point of view, it is not possible that in the same plane some bearing contact zones have wears and others do not. The proposed deep learning system was able to correctly classify all components, thus achieving an accuracy of 100%. Clearly, the inspection based on a YOLOv5+DeepLabV3+ ensemble is effective for the classification of remanufactured cages and that the decision tree is a necessary algorithm in the final decision making.

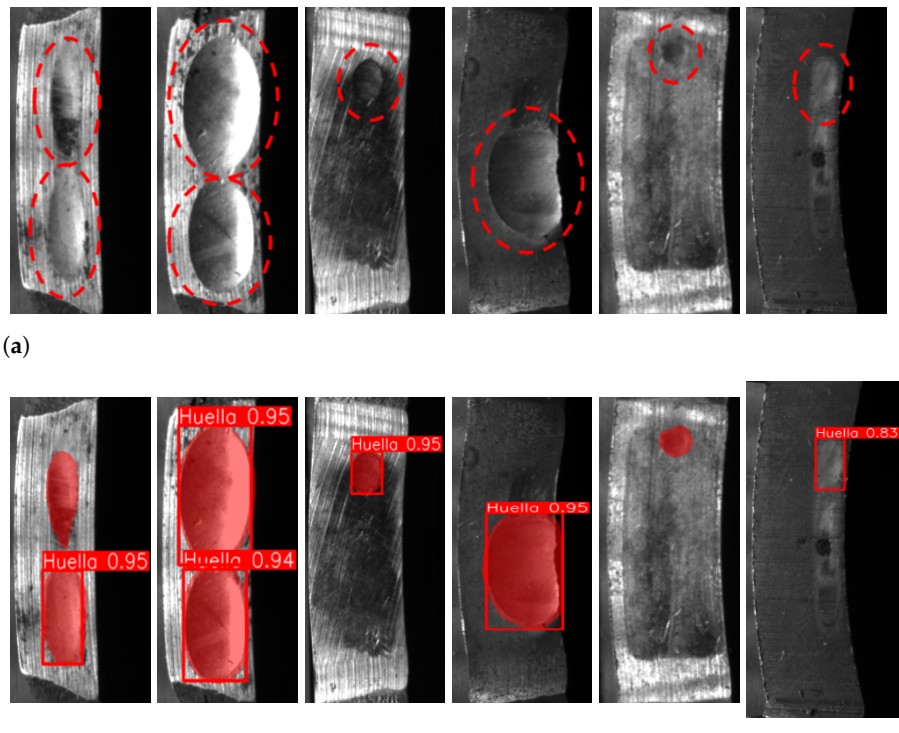

(**a**)

(**b**)

**Figure 9.** Surface wear detection by ensemble DeepLabV3+ and YOLOv5 models; (**a**) CV joint cage bearing contact point images and the wear zone indicated by red dotted lines, and (**b**) predictions in red of semantic segmentation mask and detection bounding box.

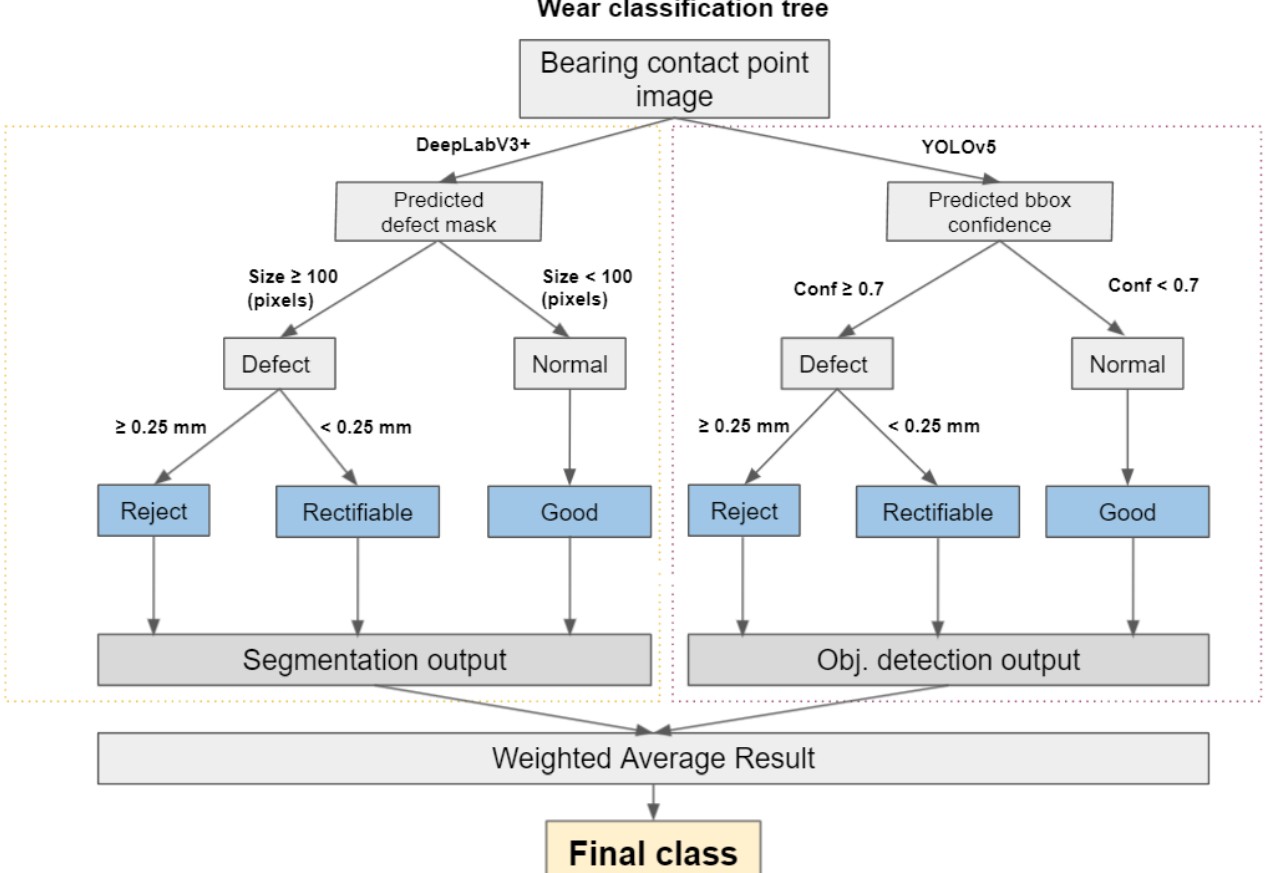

**Figure 10.** Wear classification decision tree.

*3.6. Results Summary*

Based on the experimentation, it can be concluded that the combination of the models provides stability and reliability in the defect detection in the components. Thanks to the previous analysis of the traditional machine learning methods and the different segmentation and object detection networks, it was possible to choose the best combination of models. The results obtained with traditional methods are not accurate enough regarding the manufacturing industry requirements. These methods detect more false positives, increasing false rejections. An automatic system based on these techniques would be unreliable and would generate more financial and environmental costs than manual inspection.

Proposed system based on the combination of YOLOv5 and DeepLabV3+, i.e., ensemble, classifies the component based on the individual results per bearing contact point region and following the customer's criteria. This system achieves an accuracy of 100% in the overall performance test, entailing a promising tool to solve the problem presented by the customer.

## 4. Conclusions

This work proposes an automatic inspection and classification system for automotive components, using model ensemble. An inspection pipeline is proposed, which allows to make decisions based on different criteria of component acceptance or rejection established by the manufacturer.

It is demonstrated that deep learning-based algorithms are able to learn complex geometries that traditional algorithms are not able to cope with. The models DeepLabV3+ and YOLOv5 are both well suited to segment wears, however, each model individually has its shortcomings. We validated how both models assembled are able to overcome the deficiencies in terms of FP and FN, thus obtaining a more robust detection at the inspection of each bearing contact point.

In order to perform a classification based on the models predictions and the defined quality criteria, a final decision layer based on a decision tree is proposed. This decision layer allows to consider the benefits of each model individually and unifies the classification as a single output.

A validation of the proposed system was carried out on a set of cages, where a 100% success rate in classification was obtained. As a future line of work, it is proposed to validate the system with a more significant set of samples to support the achieved results.

**Author Contributions:** Data curation, G.A.; Formal analysis, F.A.S.; Methodology, F.A.S.; Project administration, F.A.S.; Software, G.A.; Supervision, I.B.; Writing—original draft, F.A.S.; Writing—review & editing, I.B. All authors have read and agreed to the published version of the manuscript.

**Funding:** This research was funded by Ihobe 2020 grant of the Basque Government.

**Institutional Review Board Statement:** Not applicable.

**Informed Consent Statement:** Not applicable.

**Data Availability Statement:** Not available.

**Acknowledgments:** We would like to thank GKN Driveline Carcastillo for letting us publish the results obtained in the performed studies with its components.

**Conflicts of Interest:** The funders had no role in the design of the study; in the collection, analyses, or interpretation of data; in the writing of the manuscript, or in the decision to publish the results.

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
