# Peer review of "An Inspection and Classification System for Automotive Component Remanufacturing Industry Based on Ensemble Learning"

_information, doi:10.3390/info12120489_

Round 1

Reviewer 1 Report

The article is interesting and timely. However, the structure of the article should be improved and the article should be made valuable to a wide audience. Detailed comments are listed below.

1. Structure of Section 1 and Section 2 must be supplemented with additional subsections. Section 1 should include subsection 1.1 devoted to remanufacturing, which will start in the middle of line 15 ("According to [4], remanufacturing could be defined as"). Subsection 1.3, describing the article's aim and structure, should also be added. Section 2 must be supplemented with at least Section 1.1, including text about constant velocity joints. 

2. 81-82: "In this work we propose an inspection and classification system for constant velocity joint cages using Deep Learning model ensembling." Why was this joint chosen as the subject of research?

3. Section 3 (and Section 2, as well) should be supplemented with a lead paragraph (located between section 3 and subsection 3.1) that briefly summarizes the content. 

4. Table 1 and Table 2 should be cited close to where it is first mentioned. 

5. The paper incudes shorthands that are understable by scholars in the field, but not necessarily by a wider audience (e.g. 237-238: "However, it is prone to have false positives (FP), which leads to a decrease in detection reliability." - why FP affects detection reliability?). Please check the text and revise.

6. Results are not clearly summarized. I suggest supplementing the paper with a section that will summarize results (in not as laconic form as presented in section 4). Comparison of obtained results with results of other automated visual inspections known from the literature would be appreciated. 

7. Some grammatical and spelling errors were detected. 

Author Response

Dear Reviewer,

Thank you so much for your comments. Please find attached the responses.

Best Regards

Reviewer 2 Report

The authors present an ensemble learning-based classification system for the automotive component remanufacturing industry. Although this paper is interesting to read, the following concerns have to be addressed:

  1. The title of section 1.1 is unnecessary.
  2. It is not clear why the ensemble model is adopted. Please clarify this.
  3. The contributions need to be summarized in the Introduction. 
  4. The way in Section 2.1.2 to implement ensemble learning is not clearly presented. Please clarify this.
  5. It will be better to revise Figure 8 into Table format.

Author Response

(The authors gave the same response as above.)

Round 2

Reviewer 1 Report

The authors have addressed the reviewer's comments. In my opinion, the paper should be published.